# Numerical Studies of the Impact of Electromagnetic Field of Radiation on Valine

**DOI:** 10.3390/ma16051814

**Published:** 2023-02-22

**Authors:** Teodora Kirova, Jelena Tamuliene

**Affiliations:** 1Institute of Atomic Physics and Spectroscopy, University of Latvia, LV-1004 Riga, Latvia; 2Institute of Theoretical Physics and Astronomy, Vilnius University, 10257 Vilnius, Lithuania

**Keywords:** valine, electric field, magnetic field, Gaussian-type orbitals, fragmentation

## Abstract

We present the results of numerical calculations of the effect of an electromagnetic field of radiation on valine, and compare them to experimental results available in the literature. We specifically focus on the effects of a magnetic field of radiation, by introducing modified basis sets, which incorporate correction coefficients to the s-, p- or only the p-orbitals, following the method of anisotropic Gaussian-type orbitals. By comparing the bond length, angle, dihedral angles, and condense-to-atom-all electrons, obtained without and with the inclusion of dipole electric and magnetic fields, we concluded that, while the charge redistribution occurs due to the electric field influence, the changes in the dipole momentum projection onto the y- and z- axes are caused by the magnetic field. At the same time, the values of the dihedral angles could vary by up to 4 degrees, due to the magnetic field effects. We further show that taking into account the magnetic field in the fragmentation processes provides better fitting of the experimentally obtained spectra: thus, numerical calculations which include magnetic field effects can serve as a tool for better predictions, as well as for analysis of the experimental outcomes.

## 1. Introduction

Amino acids are the structural units of proteins. By joining together, amino acids form peptides (short polymer chains) or polypeptides/proteins (longer polymer chains). The 20 amino acids encoded directly in the universal genetic code are standard/canonical amino acids. Non-protein amino acids also have important roles as metabolic intermediates, such as in biosynthesis, or are used to synthesize other molecules. All amino acids have their specific role in the organism: for example, serine and threonine are proteinogenic amino acids, with an alcohol group in the side chains [1]. Serine plays a crucial role in the metabolism of, and signaling activities in, living organisms [2], and in the brain development of embryos [3]. Serine exists in the active sites of many enzymes, and has an essential role in their catalytic function. Threonine is an important constituent of collagen, elastine, and enamel protein. Threonine contributes to better function of the digestive tracts, and its deficiency has been associated with skin disorders and weakness. Tryptophan is the metabolic precursor of seroton [4], which is necessary for normal growth in infants, and for nitrogen balance in adults. The L-stereoisomer of tryptophan is used in structural or enzyme proteins, while the R-stereoisomer is occasionally found in naturally produced peptides. The D-isomer of the molecule is not utilized by humans [5], while L-tryptophan is used for coping with insomnia, sleep apnea, depression, anxiety, attention deficit hyperactivity disorder, and Tourette’s syndrome.

Valine is an α-amino acid that is used in protein biosynthesis, and thus is essential in humans, and must be obtained through the diet. In sickle-cell disease, valine substitutes for hydrophilic glutamic acid, and as valine is hydrophobic, the hemoglobin is prone to abnormal aggregation.

Any damage of the amino acids leads to a disorder in the living organism. One of the reasons for such damage is radiation. It is very well known that common sources of ionizing radiation are radioactive materials that emit α, β or γ radiation, the source of X-rays from medical radiography examinations, and cosmic rays. A better understanding of the damaging processes allows one to understand the nature of radiation-induced diseases, to find an effective treatment for them, and to create advanced equipment for medical examination. The interaction of molecules with radiation is a fundamental and very important process in various fields, e.g., in radiation biology. Shortly after the deposition of high-energy ionizing quanta into a biological medium [6], electrons with different energies are formed, and are able to destroy biological molecules, such as DNA and proteins, and cause chromosome aberrations, leading to cancer, mutations, genetic transformations, etc., [7]; therefore, it is extremely important to investigate in detail the underlying mechanisms of the above processes, by studying the degradation of the biosystem sub-units under electron impact. Because of their scientific and medical interest, many research groups have studied the structural changes of amino acids, using electron ionization mass spectrometry, where the mass spectra are typically interpreted by theoretical calculations [8,9,10,11]. The accuracy of such theoretical investigations is crucial to the interpretation of the experimental results.

In our study, the radiation was entered, through the inclusion of electric and magnetic fields with different strengths, into the numerical calculation: this allowed us to simulate the damage to valine under the impact of the different energy carried by an electromagnetic wave. It is well known that in an electric field the electron moves at a constant velocity normal to the field direction, but accelerates along it, while the magnetic field force is always directed at a right angle to the electron motion direction: thus, the resulting path of the electron is a circle. It is predicted that the electric field’s influence on molecule fragmentation is more significant than that of the magnetic field, because the chemical bond strength would change, due to electron density changes; however, some fragmentation processes could be slower, due to disordered electron movement caused by the magnetic field. To exhibit the influence of the magnetic field on the fragmentation process, we investigated the geometric and electronic structure of one of the most stable conformers: valine. This amino acid was chosen for our investigation due to a previous study [12], providing comparison of theoretical and experimental results on the interaction of the valine molecule with a high-energy electron beam.

The paper is organized as follows. In Section 2 we outline the basics of the electromagnetic field of radiation interacting with matter, emphasizing the role of the magnetic field; specifically, the method of anisotropic Gaussian-type orbital (AGTO) basis is introduced, as a way of incorporating the effect of the magnetic field of radiation into the molecular wave functions. Subsequently, in Section 2, we provide a detailed overview of the AGTO method, and apply the AGTO construction scheme for the H,C,N,O atoms, of which amino acids are constituted. Section 3 presents the results of the numerical calculations of the effect of the electromagnetic field of radiation on valine, where modified basis sets were used, incorporating the correction coefficients, which depended on the magnetic field of the molecular wave functions. Section 4 is dedicated to analysis of the obtained results. The paper ends with Conclusions and Acknowledgments.

## 2. Materials and Methods

### 2.1. Atoms/Molecules in Electromagnetic Fields

When accelerated particles have charges, they produce electromagnetic (EM) waves. The radiation is an irreversible flow of EM energy, from the source (charges) to infinity: this is possible only because the EM fields associated with accelerating charges have a 1/r instead of $1/r^{2}$ dependence, as is the case for charges at rest or moving uniformly. Thus, the total energy flux obtained from the Poynting flux is finite at infinity.

The radiation field at point *P* at time *t*, which is generated by a moving charge, is calculated from the retarded scalar and vector potentials V and A, using E=−∇V−∂A/∂t and B=∇×A. After some mathematical calculations, the famous retarded Liénard–Wiechart potentials for a moving point charge are obtained:(1)$$A=\frac{\mu_{0}}{4\pi }\frac{qv(t^{{}^{\prime }})}{(1-\hat{R}\cdot v\left(t^{{}^{\prime }}\right)/c)R}=\frac{v}{c^{2}}V\left(r,t\right).$$

Here, $\hat{R}$ is the unit vector between the moving charge at point *P* at the retarded time, while v is the velocity of the charge.

As there is an implicit *r*-dependence in the retarded time, differentiation of the potential leads to a 1/r-dependence in the fields: the latter results in a net flow of EM energy towards infinity.

The differentiation of the Liénard–Weichart potentials gives the radiation field of the moving charge:(2)$$\begin{array}{ccc} E(r,t) & = & \frac{q}{4\pi \epsilon_{0}}\left\{\frac{\hat{R}}{{\left(1-\hat{R}\cdot \beta \right)}^{3}R}\times \frac{1}{c}\left[\left(\hat{R}-\beta \right)\times \dot{\beta }\right]\right\},\\ B(r,t) & = & \frac{1}{c}\hat{R}\times E\left(r,t\right), \end{array}$$
where β and $\dot{\beta }$ are the charged particle’s velocity and acceleration divided by *c* at the retarded time, while E, B, and $\hat{R}$ are mutually perpendicular.

The many-electron Hamiltonian of an atom or molecule in a magnetic field can be written in terms of sums of orbital and spin angular momenta:(3)$$\hat{H}=\hat{\mathrm{Ho}}+\frac{1}{2}\sum_{i=1}^{N}B\cdot \mathrm{lo},i+B\cdot S+\frac{1}{8}\sum_{i=1}^{N}\left(B^{2}{ro,i}^{2}-{\left(B\cdot \mathrm{ro},i\right)}^{2}\right),$$
where S is the total spin, ro,i denotes the position of the *i*th electron with respect to the global gauge origin O, and $\mathrm{lo},i=-i\mathrm{ro},i\times \nabla_{i}$ is the canonical angular momentum.

Choosing the gauge origin O=0 and the magnetic field in the direction of the *z* axis, e.g., $B=B\hat{z}$, the above equation simplifies to:(4)$$H=H_{0}+\frac{1}{2}BL_{z}+BS_{z}+\frac{1}{8}B^{2}\left(x^{2}+y^{2}\right).$$

In this case, only the *z* projections of the orbital and spin angular momentum operators $L_{z}$ and $S_{z}$ contribute to the orbital-Zeeman and spin-Zeeman terms. For a more detailed description, the reader is referred to [13].

Applying an EM field means that the atoms/molecules are perturbed by both the electric and magnetic fields. The typical energy shift in the linear Stark effect is $\Delta E∼e\varepsilon a_{0}∼e\varepsilon \hbar /mc\alpha$, while in the Zeeman effect it is ΔE∼eBℏ/2m∼eϵℏ/2mc; therefore, the effects of the electric field are larger by a factor of 1/α (α=1/137 is the fine structure constant).

For this reason, as a first approximation in our studies, we will neglect the terms of the magnetic field in the Hamiltonian, and leave only the ones including the electric field. The effects of the *B* field of the EM radiation will be accounted for only by using the method of anisotropic Gaussian-type orbital basis, introduced in the next subsection, which brings in an anisotropy in the wave function in order to describe the elongation of electron orbitals and densities along the field direction.

### 2.2. Anisotropic Gaussian-Type Orbital Basis Sets for Atoms in Intermediate Magnetic Fields

The regime of medium magnetic field strength (0≤B≤1 a.u.; 1 hartree a.u. $=2.3505\times 10^{5}T$) is challenging to calculate, because of the different symmetries of the Coulomb and Lorentz interactions, neither of which can be treated as a perturbation. Typical for this regime is the use of the so-called anisotropic Gaussian-type orbital (AGTO) basis, first introduced by Schmelcher and Cederbaum [13]. In such a basis, the exponential factor of the basis functions has different decay constants along, and perpendicular to, the direction of the magnetic field. Such an anisotropy gives the flexibility to describe the elongation of electron orbitals and densities along the field direction. In cylindrical coordinates (ρ, *z*, ϕ), the *j*th AGTO basis function takes the form:(5)$$\chi_{j}\left(\rho ,z,\phi \right)=N_{j}\rho^{n_{\rho_{j}}}z^{n_{z_{j}}}e^{-\alpha_{j}\rho^{2}-\beta_{j}z^{2}}e^{im_{j}\phi },j=1,2,3\ldots ,$$
where $n_{\rho_{j}}=\left|m_{j}\right|+2k_{j}$, $k_{j}=0,1\ldots$ with $m_{j}=\ldots -2,-1,0,1,2\ldots$ and $n_{z_{j}}=\pi_{z_{j}}+2l_{j}$, $l_{j}=0,1\ldots$, with $\pi_{z_{j}}=0,1$. The *B* field compresses the transverse (radial) functions, which are related to the exponents $\alpha_{j}$, relative to the corresponding longitudinal (axial) ones (exponents $\beta_{j}$), so that we will always have $\alpha_{j}\geq \beta_{j}$. Alhough Schmelcher and Cederbaum deduced all the required Hamiltonian matrix elements with respect to such AGTO basis functions, the optimal determination of the sets $\alpha_{j}$ and $\beta_{j}$ is, practically, an open question. Alternatively, one can use nearly optimized basis sets, as for example in [14], where Kravchenko and Liberman (KL) investigated the hydrogen atom and molecular ion, and showed that systematically constructed AGTO basis sets provide accuracy of $10^{-6}H$ or better. Zhu and Trickey constructed KL-like highly optimized basis sets without requiring case-by-case full non-linear optimization [15]. Subsequently, after detailed analysis and extensive numerical exploration [16], the same authors presented a significant technical advance of the AGTO basis sets. For light multiple-electron atoms and ions, the strong electron–electron repulsion at the vicinity of the nucleus was considered, and the basis sets were modified, leading to orbital-quantum-numbers and electron-occupation-numbers dependence, as well as dependence on the *B* field strength and nuclear charge *Z*. While the absolute basis set errors varied from a few hundredths to a few mH, the relative errors in a wide range of magnetic field strengths were similar for different atoms and ions, showing satisfactorily the accuracy of the constructed basis sets.

### 2.3. AGTO Construction Scheme

We followed the procedure outlined by Zhu and Trickey [16], in order to obtain the $\alpha_{j}$ and $\beta_{j}$ sets for each of the atoms of which the valine molecule consists.

The procedure can be summarized as follows. According to [17], the following Gaussian sequence of length $N_{b}$ and rule of formation gives a highly accurate total energy value for the hydrogen atom at zero magnetic field, e.g., B=0:(6)$$\begin{array}{ccc} \beta_{j} & = & pq^{j},j=1,2\cdots N_{b}=16,\\ \ln p & = & a\ln\left(q-1\right)+a^{\prime },\\ \ln\left(\ln q\right) & = & b\ln N_{b}+b^{\prime },\\ a & = & 0.3243,a^{\prime }=-3.6920,\\ b & = & -0.4250,b^{\prime }=0.9280. \end{array}$$

Following the above, we determined the initial parameters *p* and *q*: this way, we had the base sequence, with respect to which the rest could be constructed. Furthermore, the parameters $A(\gamma ,m,\pi_{z})$ and $D(\gamma ,m,\pi_{z})$ were introduced as:(7)$$\begin{array}{ccc} D(\gamma ,m,\pi_{z}) & = & 0.4+\frac{0.6\left(l+1\right)/\left(l^{2}+l+1\right)}{1+1.105{\left(l+1\right)}^{3}\gamma^{0.425(l+2)}},\\ A(\gamma ,m,\pi_{z}) & = & \frac{0.02073+0.00035(2\pi_{z}+l\left(l-1\right)/3)}{D{\left(\gamma ,m,\pi_{z}\right)}^{1.25}}. \end{array}$$

Here, $\gamma =B/Z^{2}$ represented the scaled magnetic field (*Z* was the nuclear charge of the atom) and $l=\left|m\right|+\pi_{z}$. The magnetic field dependence of the $\alpha_{j}$ coefficients was given by a function, $\Delta_{j}\left(\beta_{j},B\right)$:(8)$$\begin{array}{c} \Delta_{j}\left(\beta_{j},B\right)=B\{\left(\frac{1}{4}-\frac{\beta_{j}}{B}\right)[1-{\left(1-e^{-30\beta_{j}/B}\right)}^{8}]\\ +\frac{A(\gamma ,m,\pi_{z})}{{\left(\beta_{j}/B\right)}^{D(\gamma ,m,\pi_{z})}}{\left(1-e^{-30\beta_{j}/B}\right)}^{8}\}. \end{array}$$

A minimum value for $\Delta_{j}\left(B\right)$ was imposed according to:(9)$$\Delta_{min}\left(B\right)=\left\{\begin{array}{cc} 0, & \mathrm{if}\;m=\pi_{z}=0.\\ \frac{0.1562B}{1+\gamma^{-0.55}}, & \mathrm{if}\;|m|=1\;\mathrm{and}\;\pi_{z}=0. \end{array}\right.$$

The above two equations gave the transverse exponents $\alpha_{j}$ of the AGTO basis functions in an arbitrary magnetic field *B*:(10)$$\alpha_{j}=\beta_{j}+max\left(\Delta_{j}\left(\beta_{j},B\right),\Delta_{min}\left(B\right)\right).$$

An additional requirement was introduced, that the asphericities of any two adjacent basis functions differed by no more than 0.03B. The latter implied that there would be at least 8 AGTO basis functions included within the range for which the asphericity changed from zero to its maximum value, 0.25.

Finally, the iteration expressions for determination of the $\beta_{j}$ coefficients were given by:(11)$$\begin{array}{ccc} \beta_{j+1} & = & min(max\left({\Delta_{j+1}}^{-1}\left(\alpha_{j}-\beta_{j}-0.03B,B\right),\sqrt{q}\beta_{j}\right),q\beta_{j}),\\ \beta_{j-1} & = & min(max\left({\Delta_{j-1}}^{-1}\left(\alpha_{j}-\beta_{j}-0.03B,B\right),\beta_{j}/\sqrt{q}\right),\beta_{j}/q),\\ \beta_{0} & = & p. \end{array}$$

In the above, ${\Delta_{j}}^{-1}\left(\Delta_{j}\left(\beta_{j},B\right),B\right)$ denoted the solution of Equation (9) with respect to $\beta_{j}$. Thus, the procedure for AGTO basis set construction for hydrogen in an external magnetic field was completed.

In cases of many-electron atoms, several complications arise, because of the electron–electron interaction: specifically, the occupation of the 1s orbital by one or two electrons causes screening of the nuclear attractive potential, making it less effective. Outer electrons will also have an effect on the 1s electron(s); if the 1s orbital is doubly occupied, there will be strong repulsion between the electrons. In order to describe the above effects, Zhu and Trickey introduced an additional scaling factor, *f*, in Equation (10); however, for simplicity, we assumed the value of *f* to be equal to 1. In addition, we considered only the orbitals with $m=0,1,\pi_{z}=0$, as they were the most dominant ones.

Following the above-described AGTO construction scheme, we started from Equation (6), in order to find the values of *p*, *q*, and $\beta_{0}$. Subsequently, we calculated the $A(\gamma ,m,\pi_{z})$ and $D(\gamma ,m,\pi_{z})$ parameters using Equation (7), and then the parameter $\Delta_{j}\left(\beta_{j},B\right)$ in Equation (9). The quantum numbers *m* and $\pi_{z}$ determined the value of $\Delta_{min}\left(B\right)$ from Equation (9), which in turn we used to calculate $\alpha_{0}$ from Equation (10). Furthermore, with the help of Equation (11), we could find the next value in the $\beta_{j}$ sequence, namely $\beta_{1}$, after which we repeated the iterative procedure, to find also $\alpha_{1}$.

In this way, we calculated the values of $\alpha_{1}$ and $\beta_{1}$ for the H,C,O,N atoms in the electric field range E=0.3÷0.7 a.u. (we changed the electric field strength gradually by 0.1 a.u.) and its corresponding magnetic field in the range B=0.0000218875÷0.0000510708 a.u. (calculated from Equation (2)). Due to the small values of the magnetic field of the radiation, we found that $\alpha_{1}=\beta_{1}=0.0539737$, regardless of the type of atom or orbital (e.g., *s* or *p*). Furthermore, we included the above-calculated correction coefficients in the basis set, in order to perform the numerical calculations of the effects of the magnetic field of radiation on valine.

## 3. Results

The measured mass spectra of the valine molecule, upon interaction with a high-energy electron beam, are presented in [12]. The most important results obtained in that study are as follows:no new peaks were observed in the 5 and 20 kGy spectra, in comparison to the one measured without interaction with a high-energy electron;the intensities of the peaks, with respect to that of the *m*/*z* = 72 fragment (corresponding to $C_{4}H_{10}N^{+}$), significantly changed;the peak intensity of the *m*/*z* = 27–29 fragments decreased differently with increase of the irradiation dose;the intensity of the *m*/*z* = 45 fragment increased, because the $CO_{2}$ molecule joined the *H* atoms formed due to irradiation, whereas the *m*/*z* = 55 and 56 fragments resulted from the decomposition of not only the parent valine molecule, but also the $C_{4}H_{9}N$ or $C_{2}H_{3}NO_{2}$ fragments formed under irradiation.

In order to model the results of the fragmentation processes obtained in the experiment of [12], we performed a theoretical investigation of valine with the inclusion of an electric dipole field in the *x* direction (see Figure 1). The theoretical approach is described in detail in our previous publications [9,12]. We used the most stable conformer of valine, along with optimization without any symmetry constraints, e.g., where all bonds length, angles and dihedral angles are changed. We implemented the Becke’s three-parameter hybrid functional approach with non-local correlation, provided by Lee, Yang, and Parr (B3LYP) [18,19,20], cc-pVTZ basis set. The calculations were performed using the GAUSSIAN09 package [21], with and without the dipole electric field influence, adding also the magnetic field contributions. The latter was implemented by using two modified basis sets, the first one in which both the *s*- and *p*-orbitals were modified by the magnetic field, and the second one in which the correction coefficients, $\alpha_{1}$ and $\beta_{1}$, which we calculated in Section 2.3, entered only the wave functions of the *p*-orbitals. The significant difference in the shape of the valine conformer under study was foreseen at a dipole magnetic field strength of 0.3 a.u. Thus, the comparisons of the geometric parameters, such as bond length, angles, and dihedral angles, were performed at exactly this strength of the electric field. The obtained results are presented in Table 1.

In addition, we computed the dipole moment, and its projection to the *x*, *y*, and *z* axes, without and with the inclusion of the dipole electric of 0.3 a.u. and the corresponding magnetic field (shown in Table 2). Table 3 summarizes the results of the geometry optimization of the one of the most stable conformers of valine, with inclusion of the electric and EM fields. The marks of the atoms used are given in Figure 1.

## 4. Discussion

Referring to the obtained results, we may state that the shape of the valine molecule under irradiation could be different in comparison to that of the non-irradiated molecule, due to the -$NH_{2}$ and -$CHO_{2}$ groups’ rotation in respect of the core of the molecule. In addition, we observed changes in the bond angle, which led to alteration of the electronic structure (as seen in Table 1 and Table 2), and explained the increasing intensities of the peaks, with respect to that of the *m*/*z* = 72 (corresponding to the fragment $C_{4}H_{10}N_{+}$). Two important observations should be made here: firstly, the different geometric shape indicated the creation of other conformers of valine; secondly, the condense-to-atom-all electrons (or bond order) between the C2–C11 and C2–C4 atoms decreased with the inclusion of the electric dipole field, which indicated the weakening of some bonds, and a larger probability of their destruction. The latter was observed both in the cases when the effects of the magnetic field were not included and in the cases when they were included in our calculations.

The magnetic field influence is also clearly seen in the obtained results. Although at first sight, the inclusion of the magnetic field effects did not seem to influence the electronic structure, the data in Table 1 shows that the values of the dihedral angles could vary up to 4 degrees when the magnetic field effects were taken into account.

In addition, the magnetic field effects could be crucial due to the alteration of the dipole momentum projection, which the field caused. The results presented in Table 2 show that in the EM field the polarity of the valine molecule increased significantly, around 4.5 times. We conclude that the latter occurred due to charge redistribution, as the bond lengths of the molecule—and, as a consequence, the charge separation—were not changed. There is no doubt that the charge redistribution occurred due to the electric field influence, as indicated by the values of the dipole momentum projection onto the *x* axis. At the same time, we observed that the values of the dipole momentum projection onto the *y* and *z* axes were also significantly different (compared to those of the non-irradiated molecules), and we concluded that these changes were related to the influence of the magnetic field.

As Table 3 shows, the effects of the magnetic field became more important with the increase of the electric dipole field strength. Firstly, the decomposition of the valine occurred at a lower strength of the dipole electric field. Secondly, the fragments formed with and without the inclusion of the magnetic field effects were different: for example, analyzing the obtained results revealed that the formation of the CHO${}_{2}$ fragment occurred only when the magnetic field effects were included in the basis set of the *p*-orbitals, an effect which was not seen when only the influence of the electric field was taken into account. According to the experimental studies of [12], the intensity of the *m*/*z* = 45 fragment increases in the mass spectra of the valine, which is explained as the CO${}_{2}$ fragment bonding to the *H* atoms formed due to the irradiation: thus, we see that the numerical results obtained in our studies, with inclusion of the magnetic field effects, allowed us to provide better fitting of the experimentally obtained spectra. The latter emphasized the importance of including magnetic field effects in future studies, as a tool to make better predictions of the experimental outcomes.

## 5. Conclusions

In this paper, we investigated theoretically and numerically the effect of the electromagnetic field of radiation on the valine molecule, which was further compared to the previously existing experimental results for the interaction of valine with a high-energy electron beam. The study included geometry optimization of the structures of valine, using the B3LYP/cc-pVTZ approach. The theoretical calculations were performed with the help of the quantum computational GAUSSIAN09 program package. Furthermore, we applied the method of anisotropic Gaussian-type orbitals, as a way to incorporate the effect of the magnetic field of radiation in the molecular wave functions. We proceeded, using the modified cc-pVTZ basis, in which the correction coefficients $\alpha_{1}$ and $\beta_{1}$ were used for all the atom types, e.g., H,C,N,O, of which the valine molecule consists. The correction coefficients were introduced to the *s*-, *p*- or only the *p*-orbitals, and a comparison of the bond length, angle, dihedral angles, and condense-to-atom-all electrons was performed without and with the inclusion of the dipole electric and magnetic fields. The magnetic field influence was clearly seen, showing that the values of the dihedral angles could vary by up to 4 degrees when magnetic field effects were taken into account. While the charge redistribution occurred due to the electric field influence, the changes in the dipole momentum projection onto the *y*- and *z*- axes were caused by the magnetic field effects. The influence of the magnetic field was also crucial in the fragmentation processes, proving that the fragments formed with the inclusion of the magnetic field were different to those formed without it (e.g., the $CHO_{2}$ fragment was seen only when the magnetic field effects were included in the basis set of the *p*-orbitals). The latter indicates that our numerical calculations, which took into account the influence of the magnetic field of radiation, provided a better fit with the experimental data, and that they can be used as a tool for more accurate predictions and analysis of present and future experimental outcomes. 

## Figures and Tables

**Figure 1 materials-16-01814-f001:**
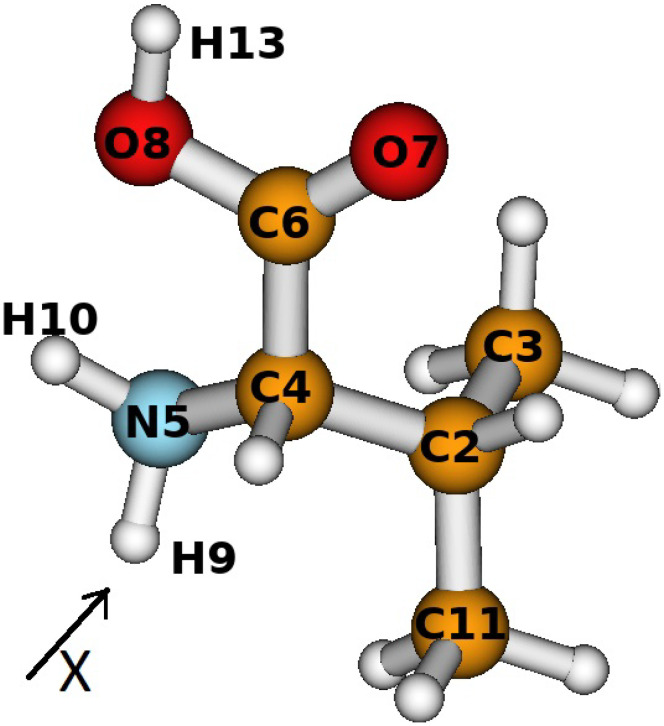
The view of one of the most stable valine conformers obtained by the B3LYP/cc-pVTZ approach. The number mark of each atom is used in the discussion below.

**Table 1 materials-16-01814-t001:** Bond length, angle, dihedral angles, and condense-to-atom-all electrons obtained without and with the inclusion of dipole electric (E) and magnetic (M) fields. The influence of the M field was added in s, p or only p-orbitals. The strength of the dipole electric field was equal to 0.3 a.u. The values in bold are for emphasis.

Bond Length, A				
**Bond**	**Valine**	**Valine in E Field**	**Valine in EM Field** **(s, p Orbitals)**	**Valine in EM Field** **(p Orbitals)**
C2-C11	1.53	1.54	1.53	1.53
C2-C3	1.53	1.53	1.53	1.53
C2-C4	1.55	1.55	1.55	1.54
C4-C6	1.52	1.56	1.55	1.55
C4-N5	1.46	1.47	1.47	1.47
C6-O7	1.20	1.21	1.20	1.20
C6-O8	1.36	1.31	1.31	1.31
N5-H9	1.01	1.02	1.02	1.02
N5-H10	1.01	1.02	1.02	1.02
O8-H13	0.97	0.98	0.98	0.98
**Condense-to-Atom-All Electrons (Bond Order)**				
**Bond**	**Valine**	**Valine in E Field**	**Valine in EM Field** **(s, p Orbitals)**	**Valine in EM Field** **(p Orbitals)**
C2-C11	0.680	**0.416**	**0.418**	**0.418**
C2-C3	0.638	0.642	0.642	0.642
C2-C4	0.554	**0.300**	**0.296**	**0.296**
C4-C6	0.442	0.586	0.576	0.576
C4-N5	0.530	0.502	0.492	0.492
C6-O7	1.566	1.458	1.468	1.468
C6-O8	0.898	1.008	1.008	1.008
N5-H9	0.718	0.744	0.742	0.742
N5-H10	0.716	0.772	0.778	0.778
O8-H13	0.630	0.668	0.672	0.672
**Bond Angle Degree, Degrees**				
**Angle**	**Valine**	**Valine in E Field**	**Valine in EM Field** **(s, p Orbitals)**	**Valine in EM Field** **(p Orbitals)**
C3-C2-C11	111.333	110.914	111.032	111.383
C11-C2-C4	110.463	110.737	110.068	110.489
C3-C2-C4	112.481	112.614	112.325	112.183
C2-C4-C6	110.862	109.778	110.093	109.046
C2-C4-N5	111.925	111.760	111.162	111.967
C4-C6-O7	**154.415**	**121.109**	**121.195**	**120.897**
C4-C6-O8	113.037	110.454	110.108	109.871
O7-C6-O8	121.984	128.428	128.694	129.232
C6-O8-H13	**106.546**	**114.158**	**113.803**	**112.937**
C4-N5-H10	111.020	108.986	108.335	107.031
C4-N5-H9	110.244	107.918	107.447	112.937
**Dihedral Angle Degree, Degrees**				
**Angle**	**Valine**	**Valine in E field**	**Valine in EM Field** **(s, p Orbitals)**	**Valine in EM Field** **(p Orbitals)**
C3-C2-C11-H17	60.321	51.561	50.052	49.582
C11-C2-C4-N5	−64.581	−65.939	−64.983	−64.276
C3-C2-C4-N5	60.493	58.895	59.284	60.662
C2-C4-N5-H9	73.831	**158.145**	**158.979**	**160.783**
C2-C4-C6-O7	−34.575	**−10.922**	**−6.738**	**−15.142**
C2-C4-C6-O8	**148.704**	**170.052**	**173.869**	**164.766**
C4-C6-O8-H13	177.167	176.189	177.042	176.120
H9-N5-C4-C2	**73.831**	**158.147**	**158.979**	**160.783**
H10-N5-C4-C2	**−166.996**	**−91.199**	**−91.500**	**−93.784**

**Table 2 materials-16-01814-t002:** The dipole moment and its projection to the *x*, *y*, and *z* axes, calculated without and with the inclusion of the dipole electric of 0.3 a.u. and magnetic fields. The values in bold are for emphasis.

Dipole Moment, Debye	x	y	z	Total
Valine	−1.09	1.064	−1.314	2.0097
Valine in E field	−8.833	1.630	0.191	8.985
Valine in EM field (s, p orbitals)	−8.872	1.629	**0.146**	9.022
Valine in EM field (p orbitals)	−8.915	**1.503**	0.194	9.043

**Table 3 materials-16-01814-t003:** The results of the geometry optimization of the one of the most stable conformers, with the inclusion of the electric and magnetic fields. The values in bold are to emphasize that the equilibrium point was not obtained, due to the presence of a saddle point, and that the molecule oscillated between two equilibrium points (the total energy is given), and tended to be degraded.

E Field, a.u.	Valine in E Field, Energy, a.u.	Valine in EM Field (s, p Orbitals), Energy, a.u.	Valine in EM Field (p Orbitals), Energy, a.u.
0.3	stable	stable	stable
	−402.586	-402.562	−402.546
0.4	unstable	unstable	unstable
	**−402.629**	**−402.600**	**−402.586**
0.5	unstable	unstable	CO${}_{2}$, $CH_{3}$, $H,C_{3}$H${}_{7}N$
	**−402.662**	**−402.650**	
0.6	CO${}_{2}$, $2H,C_{4}$H${}_{9}N,$	CO${}_{2}$, $3H,C_{4}$H${}_{8}N$	CHO${}_{2}$, $CH_{2}$, $H,C_{3}$H${}_{7}N$
	CO${}_{2}$, $3H,C_{4}$H${}_{8}N$		
0.7		C, 2O, $2H,C_{4}$H${}_{9}N$	CO${}_{2}$, $CH_{2}$, $2H,C_{3}$H${}_{7}N$

## Data Availability

The data presented in this study are available on request from the corresponding author. The data are not publicly available, due to authors’ rights.

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
