# Peer review of "Numerical Studies of the Impact of Electromagnetic Field of Radiation on Valine"

_materials, 2023, doi:10.3390/ma16051814_

Round 1

Reviewer 1 Report

In their manuscript, the authors present the results of electronic structure calculations to predict the response of valine to electric and magnetic fields, which may be caused by radiation. Anisotropic Gaussian Type Orbitals are employed to approximate the magnetic field effect on the molecule, and an influence on the degradation products is found when these are only applied to the p-orbitals.

In the current form, I would not recommend the publication of the manuscript, because in my opinion there are some major open questions that should be addressed:

1. More information about the underlying assumptions about the radiation is needed to fully understand the manuscript:
    - Where does such radiation come from? How do we benefit from a better understanding of the damaging process?
    - How does the radiation enter the simulation? Only through the electric and magnetic fields? Is this enough to describe radiation damage? Please discuss a bit more.

2. The Methods are not described sufficiently:
    - How does the electric field enter the Hamiltonian?
    - In which directions do you apply the electric and magnetic field? Why did you choose that particlular direction and how would a different direction influence your results?
    - How did you determine the values of the electric and magnetic fields that you apply in the simulation? (l. 186 tells me some value was "foreseen", which tells me you must have had some reasoning behind it)
    - Why does it make sense to apply the magnetic field only to the p-orbitals?

3. I wasn't able to understand the AGTO construction scheme based on the text. In particular I wasn't able to answer these questions based on the manuscript:
    - If a, b, a', b', and N_b are constants, then p and q are constants as well. Why is equation (6) so complex then?
    - If we start at beta_0 and then iteratively calculate for j>0, why is beta_(j-1) needed?

4. The results and discussion sections need to be improved to better understand what's really going on, here are some suggestions:
    - You could plot the Charge Density Difference with and without the EM fields to visualize the change in electron density / bond strength
    - You could model a few more molecules to prove that the inclusion of B-fields are also relevant in other amino acids (I don't know how long these specific calculations took, but usually a few molecules should not be too costly, I think)
    - What does the "unstable" in Table 3 mean in detail?
    - Why do you get different degradation products when including the B-field, even though the effect on most of the properties that you show is very small?

If these points are addressed, this could be a very interesting and insightful computational study that would be of interest to everyone modelling electric and magnetic field-effects in amino acids and other molecules.

Author Response

Dear Reviewer,

The authors would like to thank you for a thorough reading of the manuscript and the comments made. Your suggestions are very helpful in improving the manuscript’s quality. We present our replies to each of them separately as listed below, while the main changes are marked by blue in the revised version of the paper.

We hope that the revised manuscript is acceptable for publication.

Your sincere,

Jelena Tamuliene

  1. More information about the underlying assumptions about the radiation is needed to fully understand the manuscript:
    - Where does such radiation come from? How do we benefit from a better understanding of the damaging process?

We consider this remark of the reviewer and have added to our introduction the following:

“It is very well known that a common source of ionizing radiation is radioactive materials that emit α, β, or γ radiation, the source of X-rays from medical radiography examinations, and cosmic rays. A better understanding of damaging processes allows one to understand the nature of radiation-induced- diseases, find an effective treatment for them, and create advanced equipment for medical examination.”   

    - How does the radiation enter the simulation? Only through the electric and magnetic fields? Is this enough to describe radiation damage? Please discuss a bit more.

As it is mentioned in our paper, the radiation entered through the inclusion of electric and magnetic fields into the calculation that strength is different. It is enough because radiation is the emission or transmission of energy in the form of electromagnetic waves consisting of oscillating electric and magnetic fields that are perpendicular to each other. The different strengths of the electric and magnetic fields allow us to simulate different energy of the electromagnetic field.

Based on this remark, in the description of the introduction, we have added:

“In our study, the radiation was entered through the inclusion of electric and magnetic fields with different strengths into the numerical calculation. This allows us to simulate the damage of valine under the impact of the different energy carried by an electromagnetic wave.”

  1. The Methods are not described sufficiently:
    - How does the electric field enter the Hamiltonian?

We used Gaussian09 program for the simulation in which the field is either involved electric multipoles (through hexadecapoles) or a Fermi contact term.

    - In which directions do you apply the electric and magnetic field? Why did you choose that particlular direction and how would a different direction influence your results?

As it is mentioned in our paper, an electric dipole field is applied in the X direction along with geometry optimization of the molecule. It means, that a molecule placed in an electric field such, that its energy is the lowest, i.e. the most probable cases are investigated, and these cases are electric field direction independent.  

    - How did you determine the values of the electric and magnetic fields that you apply in the simulation? (l. 186 tells me some value was "foreseen", which tells me you must have had some reasoning behind it).

The values of the electric field strength are changed gradually by us.  To avoid misunderstanding, on line 151 of the manuscript (line 159 of the revised version) we have added:

“We change the electric field strength gradually by 0.1a.u.”

    - Why does it make sense to apply the magnetic field only to the p-orbitals?

We would like to pay attention, that the magnetic field was applied for s and p orbitals since they are the most dominant ones (see line 142 (new line 150 in the revised manuscript)).

  1. I wasn't able to understand the AGTO construction scheme based on the text. In particular I wasn't able to answer these questions based on the manuscript:
    - If a, b, a', b', and N_b are constants, then p and q are constants as well. Why is equation (6) so complex then?
    - If we start at beta_0 and then iteratively calculate for j>0, why is beta_(j-1) needed?

We thank the reviewer for these questions. The forms of Eqs. 6 and 11 were taken from reference [16] where Zhu and Trickey describe their AGTO construction scheme. However, the authors do not provide much explanation as to the exact form of these equations. They mention that they have taken the form of Eq. 6 from [17], since this exact form gives “a very highly accurate total energy for the non-relativistic H atom at B = 0 a.u”.

The form of Eq. 11 has been derived by Zhu and Trickey themselves, based on the assumption that “we require the asphericities of any two adjacent basis functions, say the jth and (j + 1)th, to differ by no more than 0.03B” and “to avoid approximate linear dependencies from excessively dense spacing of functions, we also require that the ratio of the longitudinal exponents of any two adjacent basis functions be no less than √ q”.

In our paper, we have provided the reader with the full form of Eq. 11, as derived by Zhu and Trickey, however, in our calculations we needed to use just the form of β0 and β1.

  1. The results and discussion sections need to be improved to better understand what's really going on, here are some suggestions:
    - You could plot the Charge Density Difference with and without the EM fields to visualize the change in electron density / bond strength.

We do not have the possibility to plot the Charge Density difference with and without EM. In our opinion, the dipole momentum better indicates a change in electron/bond strength. Because dipole momentum is a measure of the separation of positive and negative electric charges within a system, that is, a measure of the system's overall polarity. These results are given in the paper.

    - You could model a few more molecules to prove that the inclusion of B-fields is also relevant in other amino acids (I don't know how long these specific calculations took, but usually a few molecules should not be too costly, I think)

Thank you for the idea for our further investigation. We would like to inform you that we have proven the significance of the magnetic field inclusion in our recent publication: Jelena Tamuliene, Teodora Kirova, Liudmila Romanova, Vasyl Vukstich, and Alexander Snegursky” Fragmentation of tyrosine by high-energy electron impact’ that is accepted for publication in EPJD, 2023 Eur. Phys. J. D _#####################_https://doi.org/10.1140/epjd/s10053-023-00594-9.

    - What does the "unstable" in Table 3 mean in detail?

As it is mentioned in the paper, “unstable” means that there is a saddle point and the molecule oscillates between two equilibrium points and tends to be degraded.

This more detailed explanation is included in Table 3 in the revised manuscript.

    - Why do you get different degradation products when including the B-field, even though the effect on most of the properties that you show is very small?

The different degradation products were received because B-field was included differently either in p or in both s and p orbitals. The results clearly indicate that the B-field inclusion is important in the study of radiation damage theoretically.

Reviewer 2 Report

Thank you so much for such a great work. It was very interesting reading your manuscript.  My question is that how significant is the difference in dihedrals ( four degrees) with and without the field? The other conformes  that you obtain, could you provide rotational isomerization for one of the dihedrals?

Author Response

9Dear Reviewer,

The authors would like to thank the referees for a thorough reading of the manuscript and the comments made. Your suggestions are very helpful in improving the manuscript’s quality. We present our replies to each of them separately as listed below, while the main changes are marked in blue in the revised version of the paper. We hope that the revised manuscript is acceptable for publication.

 Yours sincerely,

Jelena Tamuliene

How is significant is the difference in dihedrals ( four degrees) with and without the field?

The difference in dihedral angles of less than four degrees is not significant without the presence of electric and magnetic fields.  Such differences in dihedrals could occur due to different basis set usage. However,  the  above difference up to 4 degrees in the presence of field is significant

 The other conformes  that you obtain, could you provide rotational isomerization for one of the dihedrals?

Our study on the Valine isomers fragmentation you may find in our paper J. Tamuliene et al. “ On the influence of low-energy ionizing radiation on the amino acid molecule: Valine case”, July 2018, Lithuanian Journal of Physics 58(2) DOI: 10.3952/physics.v58i2.3743. In this paper we presented the views of the isomers and their energy. 

Reviewer 3 Report

The authors did a computational study about the impact of electromagnetic field of radiation on molecule valine. One of the approaches to incorporate magnetic field into electronic Hamiltonian, AGTO basis set, is used in their research. The authors introduced their AGTO construction process in details. Finally, they used their new schemes to calculate the valine molecule. 

I don't think this manuscript should be accepted to be published on Materials. I have the following reasons and suggestions.

1. Your conclusion is about the accuracy of the theoretical method compared with experimental results. However, I believed that the accuracy of AGTO has already been researched systemically. This is about why you want to publish this research. What the authors can learn from you?   (https://aip.scitation.org/doi/10.1063/1.5004713) 

2. How did you incorporate AGTO magnetic field into electron Hamiltonian? Did you modify the source code of Gaussian09? Did you introduce the details in your citation 9,12?

3. I agree that magnetic field has less impact on properties of a molecule compared with Coulomb field. But it does play an important role in mass spectrometry fragmentation processes. I think you can research the decomposition pathway in of valine in magnetic field, rather than calculate the single molecule. 

4. Your research is far away from the topics of journal Materials. 

Author Response

Dear Reviewer,

The authors would like to thank you for a thorough reading of the manuscript and the comments made. Your suggestions are very helpful in improving the manuscript’s quality. We present our replies to each of them separately as listed below, while the main changes are marked in blue in the revised version of the paper. We hope that the revised manuscript is acceptable for publication.

 Yours sincerely,
Jelena Tamuliene

  1. Your conclusion is about the accuracy of the theoretical method compared with experimental results. However, I believed that the accuracy of AGTO has already been researched systemically. This is about why you want to publish this research. What the authors can learn from you?   (https://aip.scitation.org/doi/10.1063/1.5004713) 

 We agree that the accuracy of AGTO has already been researched systematically. We applied this method to include magnetic field influence to model the degradation of valine under radiation impacts. The readers can learn from us that not only electric but the magnetic field must also be included in the simulation of radiation, the amino acid degradation results are dependent on which orbital(s) magnetic field is included, and it is not necessary to change Hamiltonian to include magnetic field.

  1. How did you incorporate AGTO magnetic field into electron Hamiltonian? Did you modify the source code of Gaussian09? Did you introduce the details in your citation 9,12?

We use AGTO method because it is allowed us to create basis functions adapted to the treatment of atoms and molecules in arbitrarily strong magnetic fields. So, it is not necessary to change the Hamiltonian and modify the Gaussian09 code. The detail on magnetic field inclusion in our cited 9, and 12 references are not introduced. The simulation is performed with the inclusion of an electric field based on the presumption that the electric field influence is stronger than that of a magnetic one.

  1. I agree that magnetic field has less impact on properties of a molecule compared with Coulomb field. But it does play an important role in mass spectrometry fragmentation processes. I think you can research the decomposition pathway in of valine in magnetic field, rather than calculate the single molecule. 

We would like to pay attention that the decomposition pathway of valine is investigated in both electric and magnetic fields. The single molecule calculation is presented to show how the electronic structure of this molecule is changed in the presence of Electric and electromagnetic fields.

  1. Your research is far away from the topics of journal Materials. 

We do not have a reply to this comment, but please consider, that we have received an invitation from the editor to submit our paper to Materials, and the title of the manuscript as well as the abstract was approved before submission. We hope, that the results of our study will be used to improve the quality of research of papers that is to become the topic of the journal Materials. 

Reviewer 4 Report

The manuscript "Numerical studies of the impact of electromagnetic field of radiation on valine" is devoted to the effects of electromagnetic field on valine using modified basis sets with the inclusion of corrections coefficients to the s-, r- or only p-orbitales  following the method of Anisotropic Gaussian Type Orbitales. 

As the main result of manuscript can be called the conclusion that the influence of a magnetic field can lead to a variation in the values of dihedral angles up to 4 degrees.

Manuscript can be accepted for publication after minor revision.

I recommend:

-paying more attention to the frequency dependence in the manuscript the effect of the influence of the magnetic field on the structure of valine;

-to present an assessment of the numerical stability of results obtained in the manuscript.

Author Response

Dear Reviewer,

The authors would like to thank you for a thorough reading of the manuscript and the comments made. Your suggestions are very helpful in improving the manuscript’s quality. We present our replies to each of them separately as listed below, while the main changes are marked in blue in the revised version of the paper.

We hope that the revised manuscript is acceptable for publication in your edited journal.

Yours sincerely,

Jelena Tamuliene.

-Paying more attention to the frequency dependence in the manuscript the effect of the influence of the magnetic field on the structure of valine;

The effects of the influence of the magnetic field on the structure of Valine are represented in Table 1.

-to present an assessment of the numerical stability of results obtained in the manuscript.

To present an assessment of the numerical stability of the results obtained the values of the total energy are added in Table 3 and a short explanation of it is given:

"Table 3. ... The values in bold are for emphasis that the equilibrium point is not
obtained due to the presence of a saddle point and the molecule oscillates between two equilibrium points (the total energy is given) and tends to be degraded."

Round 2

Reviewer 1 Report

Dear authors,

Thank you for replying to the questions I had, and taking my feedback into account for improving your manuscript. I think I fully understand the manuscript now. Nevertheless, there are still two points that concern me:

1. You model radiation damage purely with polarizing effects of electric and magnetic fields. But ionizing radiation can also simply nudge an electron out of the molecule to ionize it. So why are there no computations of ionized molecules in your study? If this has been done already for this molecule, you should mention it and cite the relevant work.

2. Just publishing the results molecule by molecule without any further higher-level insights is not a good practice in my opinion. I would have preferred, if you at least tried to understand why it makes such a big difference whether the B-field is applied to the s- and p-orbitals or just the p-orbitals, and/or compared to other existing work to identify common trends or mechanisms. However, it's not up to me, but up to the editor to decide if this is okay with the guidelines of mdpi materials.

Author Response

Dear Reviewer,

Thank you for your comments. They are very helpful for our future investigations and scientific development. We present our replies to each comment separately. Please see below.

With respect,

Jelena Tamuliene

  1. You model radiation damage purely with polarizing effects of electric and magnetic fields. But ionizing radiation can also simply nudge an electron out of the molecule to ionize it. So why are there no computations of ionized molecules in your study? If this has been done already for this molecule, you should mention it and cite the relevant work.

    Thank you for this remark. Yes, of course, ionizing radiation can also simply nudge an electron out of the molecule to ionize it.  The investigation of the ionized valine was performed and published at https://www.lmaleidykla.lt/ojs/index.php/physics/article/view/3743/2541
    But the results are not necessary for our study - the ionization energy computed with and without the inclusion of a magnetic field should coincide with the experimental measurements of the parameter.

    2. Just publishing the results molecule by molecule without any further higher-level insights is not a good practice in my opinion. I would have preferred, if you at least tried to understand why it makes such a big difference whether the B-field is applied to the s- and p-orbitals or just the p-orbitals, and/or compared to other existing work to identify common trends or mechanisms. However, it's not up to me, but up to the editor to decide if this is okay with the guidelines of mdpi materials.

     We agree with your opinion that higher-level insights must be published.  In our case - the higher level is the inclusion of a magnetic field in the investigation of the fragmentation due to radiation of amino acids. But we do not think, that there is a need for any explanations concerning the difference in whether the B-field is applied to the s- and p-orbitals or just the p-orbitals, because the reasons for that are very well known - the usage of different basis sets as well as approach leads to different results in the investigation. In addition, to our knowledge, there are no results of the investigation of fragmentation of valine to be compared with ours, although we prove that the inclusion of magnetic field gives better coincidence among theoretical and experimental results in the case of tyrosine ( see Eur. Phys. J. D (2023) 77:13 
    https://doi.org/10.1140/epjd/s10053-023-00594-9.

Reviewer 3 Report

This version is improved after reversion.

Round 3

Reviewer 1 Report

Dear authors and editor,

I accept that you don't want to include any further research in this publication. So I agree for this to be published, even though I still think that a study with further insights would be of much higher interest to the community.

Best regards